# Post-Translational Modifications of Histone Variants in the Absence and Presence of a Methionine-Depleting Enzyme in Normal and Cancer Cells

**DOI:** 10.3390/cancers15020527

**Published:** 2023-01-15

**Authors:** Serena Montalbano, Samanta Raboni, Simone Sidoli, Andrea Mozzarelli, Stefano Bettati, Annamaria Buschini

**Affiliations:** 1Department of Chemistry, Life Sciences and Environmental Sustainability, University of Parma, Parco Area delle Scienze 11/A, 43124 Parma, Italy; 2COMT (Interdepartmental Centre for Molecular and Translational Oncology), University of Parma, Parco Area delle Scienze 11/A, 43124 Parma, Italy; 3Department of Food and Drug, University of Parma, Parco Area delle Scienze 23/A, 43124 Parma, Italy; 4Institute of Biophysics, National Research Center, Area della Ricerca di Pisa, Via G. Moruzzi 1, San Cataldo, 56124 Pisa, Italy; 5Interdepartmental Center SITEIA.PARMA, University of Parma, Parco Area delle Scienze 11/A, 43124 Parma, Italy; 6Department of Biochemistry, Albert Einstein College of Medicine, Bronx, NY 10461, USA; 7Department of Medicine and Surgery, University of Parma, Via Gramsci 14, 43126 Parma, Italy

**Keywords:** methionine depletion, histone variants, HT-29 cell line, Hs27 cell line, methionine gamma-lyase, mass spectrometry

## Abstract

**Simple Summary:**

Cancer cells exhibit unique metabolic properties, including a high requirement for methionine. However, the mechanism that explains how cancer cells suffer from the absence of methionine more significantly than healthy cells remains elusive. Methionine is essential for epigenetic reprogramming of cells, both at the DNA level and for post-translation histone code modification. The post-translational modifications of histone variants (hPTMs) in normal and cancer cells were characterized by mass-spectrometry in the absence and presence of methionine gamma-lyase (MGL), a bacterial enzyme that degrades methionine and inhibits the growth of cancer cells. Results indicate a complex pattern of PTMs on histone variants and striking differences between normal and cancer cells that might help in the understanding of the molecular mechanisms triggered by methionine depletion and in the fine-tuning of MGL-based cancer therapy.

**Abstract:**

Methionine is an essential amino acid involved in the formation of polyamines and a precursor metabolite for DNA and protein methylation. The dependence of cancer cells on methionine has triggered extensive investigations aimed at its targeting for cancer therapy, including the exploitation as a therapeutic tool of methionine γ-lyase (MGL), a bacterial enzyme that degrades methionine, capable of inhibiting cancer cells growth due to methionine starvation. We have exploited the high-resolution power of mass spectrometry to compare the effects of reduced availability of the methyl donor SAM, induced by MGL treatment, on the post-translational modifications of the histone tails in normal Hs27 and cancer HT-29 cells. In the absence of MGL, our analysis detected a three-fold higher relative abundance of trimethylated K25 of H1.4 in HT-29 than Hs27 cells, and a complex pattern of methylated, unmethylated and acetylated peptides in H2 and H3.3. In the presence of MGL, in HT-29, the peptide H2A1_4_11 is predominantly unmodified with mono-methylated K5 increasing upon treatment, whereas in Hs27 cells, H2A1_4_11 is monomethylated at K5 and K9 with these marks decreasing upon treatment. The time dependence of the effects of MGL-mediated methionine depletion on PTMs of histone variants in HT-29 cancer cells was also monitored. Overall, our present data on histone variants H1, H2A, H2B as well as H3.3 integrated with our previous studies on histones H3 and H4, shed light on the epigenetic modifications associated with methionine starvation and associated cancer cell death.

## 1. Introduction

Malignant cells exhibit proliferative advantages when compared to non-malignant cells due to an altered metabolism that supports their rapid growth. The Warburg effect that describes the preference of cancer cells to metabolize glucose anaerobically rather than aerobically, even under normoxia, is a typical example. In addition, cancer cells often present an increased demand for selected amino acids (AAs), resulting in a dependency on exogenous sources or upregulation of their de novo synthesis [1]. The exploitation of cellular peculiarities of cancer cell metabolism represents a promising strategy to selectively hit them for cancer treatment. In this context, amino acid starvation therapy has emerged as a potential approach. It has been reported that in cancer cells, in which amino acids biosynthetic pathways may be silenced and consumption is elevated, nutrient starvation causes cell death [2]. Recently, FDA approval of asparaginase, a well-established asparagine-depleting enzyme, in the treatment of acute lymphocytic leukemia, demonstrated the power and applicability of an amino acid (nutrient) starvation strategy in cancer therapy.

A notable example of metabolic vulnerability in cancer cells is the elevated requirement for methionine [3,4]. The high frequency of methionine auxotrophy in cancer cells suggests it might be a general metabolic defect that was termed “Hoffman effect” [5]. Non-cancerous cells do not suffer from methionine restriction for their proliferation, since they possess the ability to grow on homocysteine, one of methionine metabolic precursors. This metabolic signature has been exploited for the development of a tailored anticancer therapy based on the systemic delivery of methionine γ-lyase (MGL) (EC 4.4.1.11), a bacterial enzyme that catalyzes the degradation of methionine. MGL-mediated methionine starvation causes growth inhibition in cancer cells [6,7,8,9] when the enzyme is delivered either free in solution [10,11], or bound or entrapped in different matrices [12,13,14,15]. Gene therapy based on the MGL gene was also reported [16]. MGL significantly attenuates tumor growth also in mouse models, suggesting therapeutic potential especially in recalcitrant cancers [8].

Methionine is involved in many relevant biological processes, such as protein synthesis, glutathione formation, polyamine synthesis and methyl group donation in methylation reactions. Thus, low methionine availability widely perturbs cellular mechanisms [17]. In particular, as precursor of S-adenosyl methionine (SAM), the universal methyl donor for the methylation of DNA, RNA and histones, scarcity of methionine may alter the epigenetic landscape both for nucleic acids and histones [18].

In yeast cells, the imbalance in SAM/SAH (S-adenosylhomocysteine) ratio alters histone methylation, increasing or reducing viability [19]. Yeast cho2Δ mutant cells, which have increased SAM/SAH ratios, have striking increases of H3K4me3, H3K36me3, and H3K79me3 marks [20]. In yeast, it is well documented that low levels of methionine or SAM activates cellular response programs to restore levels of methionine and its metabolites by induction of methionine transporters, upregulation of de novo methionine biosynthesis, and autophagy. Under prolonged methionine restriction, cell cycle arrest is observed (“SAM-checkpoint”), and apoptosis is induced to protect the organism from cells unable to maintain chromatin methylation marks [21].

Sensing methionine availability is thus crucial for cellular, epigenetic, and genomic integrity, but the mechanisms for methionine perception are different throughout different organisms, despite a conserved role of methionine as a growth signal.

In mammalian, the molecular mechanisms are still elusive. It was reported that methionine is sensed indirectly through SAM [22], which binds to the mTORC regulator SAMTOR and disrupts the SAMTOR-GATOR1 complex, signaling methionine sufficiency to mTORC1. In addition, methionine starvation leads cells to arrest in the G1, S, or G2 phases depending on the cancer cell line analyzed [23], but it has not yet been clearly established whether starvation-induced death is associated with apoptosis or other cellular processes such as autophagy.

A few recent studies have started to address the epigenetic changes occurring in cancer cells exposed to methionine restriction [6,24,25,26,27,28,29,30,31]. Changes in histone methylation marks, with decrease of H3K4me3, H3K9me3, and H3K9me2 levels and possible involvement of heterochromatin regulation, were documented upon starvation either in methionine-free or enzyme-deprived cell culture media [6].

Our laboratory carried out an extensive characterization of the differential histone methylome, acetylome and proteome of normal and cancer cells upon MGL-induced methionine deprivation by mass spectrometry-based proteomics [6]. MGL treatment was found to cause cytotoxicity, changes in chromatin and proteome regulation, and spurious transcription in HT-29 colon adenocarcinoma cancer cells. Furthermore, our analysis highlighted the role of selected marks on H3 and H4 histones in reducing cancer cell fitness and suggested a direct link between MGL-mediated methionine deprivation and unfolding of constitutive heterochromatin.

The chromatin of individual eukaryotic cells is wrapped around histone octamers to form the nucleosome core particles. An octamer contains two of each core histone (H2A, H2B, H3, and H4) with 145–147 base pairs of DNA wound around it. A histone known as linker histone H1 is bound to the outside of the nucleosome core particle, forming a full nucleosome, and stabilizing higher-order chromatin structures.

Histone variants diverge in sequence and impart effects on many DNA-templated processes and are deposited throughout the cell cycle often by variant-specific histone chaperones. Replacement of canonical histones by variants affects nucleosome structure and accessibility of the histone tail. Variants of histones H2A, H2B, H3, and H1 are well-described in literature, while there is little evolutionary divergence in H4.

Distinct amino-terminal modifications of the histone tails characterize a specific “histone code” that determines dynamic transitions between transcriptionally active or transcriptionally silent chromatin states.

Histone amino-terminal modifications include acetylation, methylation, phosphorylation, ubiquitinylation, sumoylation, ADP ribosylation, and deamination [32]. In particular, acetylation at lysine residues on histone tails influences the compaction state of chromatin, weakening the electrostatic interaction between negatively charged DNA and histones [33]. Histone acetylation is mainly associated with active transcription [34]. In respect of acetylation, methylation of lysine and arginine residues on histone tails represents a complex chromatin modification. Multiple methylation states exist for both lysine and arginine residues [35]; a high degree of methylation (i.e., tri-methylation) is generally related to non-transcribed chromatin [36,37]. Histone tail phosphorylation (H1 and H3) is implicated in chromosome condensation during mitosis. Serine 10 phosphorylation on H3 is related to the induction of several genes that are rapidly transcribed after specific signal, such as c-jun, c-fos and c-myc [38]. Phosphorylation of the histone gamma-H2AX on the Ser139 in the presence of DNA damage represents the first signaling step in DNA damage repair of DNA double strand breaks [39].

This process increases the information potential of the genetic code and provides the cell with a rapid response to solicitations/stresses that require modulation of transcription. However, the histone code and its role together with the relevance of PTMs on histone variants are still not completely deciphered.

To our knowledge, our present study describes for the first time how MGL-mediated methionine depletion affects post-translational modifications deposited on histone variants (hPTMs) in normal and cancer cells. Our effort was to help better understand the complex epigenetic cell alterations due to methionine restriction but also to define new biomarkers to better calibrate anticancer therapy based on enzymes able to cause methionine deprivation.

## 2. Materials and Methods

### 2.1. MGL Expression and Purification

*E. coli* BL21(DE3) cells containing the plasmid with MGL gene from *C. freundii* were grown. Enzyme was purified, and protein concentration was determined according to the protocol previously described [12,40,41]. Protein purity was above 90% as assessed by SDS-PAGE. MGL degrades methionine according to the reaction:

L-methionine + H_2_O → methanethiol + NH_3_ + α-ketobutyrate

MGL catalytic activity was determined using L-methionine as a substrate and measuring the rate of α-ketobutyrate production in the coupled reaction with D-2-hydroxyisocaproate dehydrogenase (HO-HxoDH) by monitoring the decrease of NADH absorption at 340 nm (Δε = 6220 M^−1^ cm^−1^) at 37 °C. The rate of α-ketobutyric acid production was measured in a coupled assay at 30 °C. The reaction mixture contained 100 mM potassium phosphate, 100 mM pyridoxal 5′ phosphate (PLP), 5 mM DTT, 0.2 mM NADH, pH 7.2. The amount of the enzyme that catalyzes the formation of 1.0 μmol min^−1^ of α-ketobutyrate at pH 8.0, 37 °C, was defined as one unit of enzyme activity. Kinetic parameters of MGL mutants in the α,γ-elimination reaction with L-methionine were previously reported to be k_cat_ = 7.9 s^−1^ ± 0.3; K_M_ = 0.8 mM ± 0.1; k_cat_/K_M_ = 9.7 × 10^3^ M^−1^s^−1^ [10].

For cell experiments, MGL solutions were equilibrated with Phosphate Buffered Saline (PBS), pH 7.4, and sterilized through Millipore GV 0.22 µm filters and stored at −80 °C. The methionine concentration in the medium for cell growth was quantified to be 400 μM by the afore mentioned coupled activity assay with HO-HxoDH. This concentration corresponds to half of the K_M_ calculated for MGL substrate in the standard assay conditions [10]. The consumption of methionine by MGL is completed in less than 1 h when the enzyme is present at IC_50_. As MGL reaction is irreversible and quantitative, no methionine is present in the culture medium after 1 h [6].

### 2.2. HT-29 and Hs27 Culture Conditions

The effects of methionine depletion caused by MGL treatment were tested on two human cell lines provided by ATCC: the human colon adenocarcinoma cell line HT-29 (ATCC, HTB-38) and the human skin fibroblast cell line Hs27 (ATCC, CRL1634). Cell culture conditions have been previously reported [6]: briefly, cells were grown in DMEM medium added with 100 U/mL penicillin, 100 μg/mL streptomycin, 2 mM L-glutamine and 10% (*v*/*v*) fetal bovine serum, in a humidified CO_2_ (5%) incubator at 37 °C.

### 2.3. Histone PTM Preparation and Analysis by Mass Spectrometry

Histone acid extraction and analysis by mass spectrometry were performed following our previously detailed protocol [6,42]. 2.5 × 10^5^ cells were seeded in 12-well plates, with 2 mL of DMEM complete medium and, after treatment, cell nuclei were isolated, and histones were extracted with 0.2 M H_2_SO_4_ for 2 h and precipitated with 33% trichloroacetic acid (TCA) overnight. The histone protein pellets were dissolved in 30 μL of 50 mM ammonium bicarbonate, pH 8.0. To ensure complete derivatization, two rounds of propionylation were performed mixing propionic anhydride with acetonitrile in a ratio of 1:3 (*v*/*v*) and adding to histone solutions in the ratio of 1:4 (*v*/*v*) for 20 min, at room temperature. After digestion, to derivatize peptide N-termini, the derivatization procedure was repeated twice.

Samples were desalted prior to LC-MS analysis by using C18 Stage-tips. They were then resuspended in 10 µL of 0.1% TFA and loaded onto a Dionex RSLC Ultimate 300 (Thermo Scientific, Waltham, MA, USA), coupled online with an Orbitrap Fusion Lumos (Thermo Scientific). Chromatographic separation was performed with a two-column system, consisting of a C-18 trap cartridge (300 µm ID, 5 mm length) and a picofrit analytical column (75 µm ID, 25 cm length) packed in-house with reversed-phase Repro-Sil Pur C18-AQ 3 µm resin.

To analyze the histones, peptides were separated using a 60 min gradient from 4%-30% buffer B (buffer A: 0.1% formic acid, buffer B: 80% acetonitrile + 0.1% formic acid) at a flow rate of 300 nl/min. Data were acquired using a data-independent acquisition (DIA) method. In particular, the full MS scan (300–1100 m/z) was acquired in the Orbitrap with a resolution of 120,000. MS/MS was performed by setting consecutive isolation windows of 50 m/z with an AGC target of 2 × 10^5^ and an HCD collision energy of 30.

Histone peptide data were analyzed using EpiProfile 2.0 software [42]. The raw abundance of each (un)modified peptide was obtained by performing extracted ion chromatography for each peptidoform. To obtain peptide relative abundance, and thus relative abundance of histone PTMs, the sum of all different modified forms of a histone peptide was considered as 100% and the area of the particular peptide was divided by the total area for that histone peptide in all of its modified forms. The relative ratio of two isobaric forms was estimated by averaging the ratio for each fragment ion with different mass between the two species. The resulting peptide lists generated by EpiProfile were exported to Microsoft Excel for statistical analysis. Due to the relatively limited number of replicates (N = 3, typical of experiments involving cell lines), data distribution could not be tested using e.g., the Shapiro-Wilk test. For this reason, data distribution of data points across replicates was assumed to be normal due to the nature of the sample, but it was not formally tested. To minimize the probability of false positives, we used a *t*-test heteroscedastic with two-tails (significant if *p* < 0.05).

### 2.4. Statistical Analysis and Data Availability

To assess statistical significance, we used a two-tails heteroscedastic parametric *T*-test. Significance was assumed if the obtained *p*-value was < 0.05. Prior to T-test calculation, protein abundances were log2 transformed, normalized by the average value of each sample and missing values were imputed using a normal distribution 2 standard deviations lower than the mean. The distribution of data points of replicates was assumed to be normal, but this was not formally tested.

Histone raw files are available for download at the repository Chorus (https://chorusproject.org/) at the project number 1712, accessed on 4 March 2021.

## 3. Results

In a previous study [6], we analyzed the effects of methionine depletion caused by MGL treatment on cancer (HT-29) and normal cells (Hs27) and documented the higher susceptibility of HT-29 cells with respect to Hs27 non-malignant cells. The concentration inhibiting HT-29 cell proliferation at 50% (MGL50) was 30 µg/mL at 72 h of incubation, to be compared with more than 460 µg/mL for Hs27. Here, we apply these conditions, capable of preventing cancer cell proliferation, to investigate how they affect the PTM pattern of histone variants, with the aim to complete the characterization previously carried out on the canonical H3 and H4 histone methylation, acetylation, and phosphorylation profiles [6]. First, we compare how histone modifications on H1, H2A, H2B and H3.3 change in HT-29 and Hs27 in the absence and presence of MGL added to the cell culture medium. Successively, we present the analysis of the PTMs profile of histone variants when HT-29 cells were treated with MGL for 24, 48 and 72 h.

### 3.1. Differences on PTMs of Histone Variants between Cancer and Normal Cell Lines

H1 histone variants—H1 is the linker histone, sits at the base of the nucleosome near the DNA entry and exit sites and is involved in the folding and stabilization of the 30 nm chromatin fiber [43]. In humans, H1 is a family of closely related, single-gene encoded proteins, including seven somatic subtypes (from H1.1 to H1.5, H1.0, and H1X). H1.1 to H1.5 are expressed in a replication-dependent manner, whereas H1.0 and H1X are replication-independent [44,45].

Methylated and acetylated fragments were detected, but unmodified peptides were found to be highly abundant (Appendix A). It is of note that lysine residues are frequent and closely spaced in H1 and peptide coverage is limited due to difficulties in MS detection of small fragments. The most significant difference between normal and cancer cells is the three-fold higher relative abundance of trimethylated K25 of H1.4 in HT-29 than Hs27 cells (Table 1).

H2 histone variants—The human genome contains 10 genes that encode for H2A peptides classified as H2A1 variants, six identical in sequence and four that vary in up to three of four positions [46]. In H2A, the most abundant peptides were the unmodified ones (Table 1). In particular, H2A1_4_11 unmod, H2AJ_4_11 unmod and H2A3_12_17 unmod were more abundant in HT-29 cells than in Hs27 (Table 1). On the contrary, H2A1_1_11 unmod and H2A1_12_17 unmod are more abundant in normal cells than in cancer cells. The relative abundance of the acetylated peptides H2A1_4_11 K5acK9ac, H2A1_1_11 K5ac, H2AZ_1_19 K15ac, H2A1_12_17 K13ac, H2A1_12_17 K15ac, and of the methylated H2A1_12_17 K13me1, H2A3_12_17 K15me1, together with H2A_1_88 H2A14s.HLQLAIR is greater in cancer cells than normal cells (Table 1). In contrast, normal cells present increased levels of the methylated peptides H2A3_12_17 K13me1, H2A1_4_11 K9me1, H2A1_4_11 K5me1, together with H2AZ_1_19 K7ac, and H2A_1_88 H2AZ.AGGKAGKDSGKAKTKAVSR (Table 1).

The relative abundance of H2B_1_29 1C.PEPAKSAPAPKKGSKKAVTKAQKKDGKKR and H2B_1_29 1H.PDPAKSAPAPKKGSKKAVTKAQKKDGKKR peptides decreases dramatically in HT29 cells with respect to Hs27 cells (Table 1).

H3.3 histone variants—Histone H3.3 represents a H3 variant and differs from the canonical histone H3 in residues 87, 89, 90, and 96 of the histone fold domain and in residue 31 at the N-terminal tail. These specific residues contribute to the distinct properties of variant H3.3 [47]. Indeed, residue 31 is located in the peptide_27_40, which shows differences in methylation states and peptide abundance in untreated samples (Appendix A). Indeed, in Hs27 H33_27_40 K27me1K36me3, H33_27_40 K27me1K36me2, H33_27_40 K36me3 are the prevalent marks, while in HT-29 H33_27_40 K27me2, H33_27_40 K27me3 are the more abundant (Table 1).

### 3.2. Effect of MGL-Mediated Methionine Depletion on PTMs of Histone Variants in HT-29 and Hs27 Cell Lines

H1 histone variants—In normal cells, upon treatment with MGL H1.2K33me1 decreases compared to the untreated cells, whereas the unmodified peptide increases significantly. Also, a decrease in H1.4K25me1 and an increase in the unmodified peptide was evident in the treated sample (despite not statistically significant) (Appendix A). On the contrary, methionine depletion caused by MGL treatment does not alter H1.4K25me3 in both cell lines (Appendix A).

In cancer cells the mono-methylation at K33 in H1.2 decreases after treatment, while the unmodified peptide increases. In addition, acetylation at K31 in H1.4 increases in treated cancer cells (Appendix A).

The abundance of the peptide corresponding to the N-terminal portion (1–35) of H1.4 versus H1.5 seems to be unaffected by the treatment in normal cells with H1.4 predominant over H1.5. On the contrary, in cancer cells H1.4 is predominant in untreated cells with respect to H1.5 but becomes equal upon treatment (Appendix A).

In normal cells, upon MGL50 treatment, the peptide corresponding to the 54–84 fragment of H1.1 increases with a decrease in the peptide characteristic of H1.5. In cancer cells, the increase in H1.5 is at the expense of H1.4 with no changes in H1.1 abundance (Appendix A).

H2 histone variants—In cancer cells, H2A1_4_11 is predominantly unmodified with the mono-methylation at K5 that increases upon treatment. On the contrary, in normal cells, H2A1_4_11 is monomethylated at K5 and K9 with these marks that consistently decrease upon treatment. The abundance of the unmodified peptide is similar in both normal and cancer cells with a change in the opposite direction: in Hs27 increases upon treatment, while in HT-29 decreases (Appendix A).

For the H2AJ_4_11 peptide, in cancer cells, K5me1 is the most abundant mark and increases upon treatment whereas K9 mono-methylation and acetylation marks decrease. In Hs27 cells, the most abundant marks in the untreated cells are K5me1 > K9ac > K9me1 > unmodified. Upon treatment K5me is the dominant mark associated with K9ac and the unmodified peptide and K9me1 almost disappears (Appendix A).

In H2AX_4_11 after treatment, K5me1 increases in normal and cancer cells, while K9me1 decreases (Appendix A).

In HT-29 cells, H2A1_1_11 peptide K5 is acetylated, and treatment causes a 10% decrease. In normal cells, the unmodified peptide is as abundant as the acetylated one and the treatment increases acetylation on K5 (Appendix A).

H2A1_12_17 K15ac is a dominant mark in cancer cells and almost absent in normal cells (Appendix A).

H2A3_12_17 K15me1 is significantly dominating in untreated cancer cells with a significant fraction of unmodified peptide. K13me1 abundance increases upon treatment. In normal cells, the dominant mark is K13me1 when untreated and K15me1 when treated.

H2A_1_88 H2AY.SAKAGVIFPVGR (macroH2A) skyrockets in normal cells upon treatment (Appendix A), while the increase is—modest in cancer cells.

In H2B1A, K86ac is the predominant mark in cancer cells independently of treatment, while in normal cells a consistent fraction is unmodified, and acetylation is strongly affected by treatment. However, statistical significance is not reached by these data (Appendix A).

Abundance of the variants affecting the N-terminal peptide of H2B is different in untreated normal and cancer cells and changes in response to MGL-mediated methionine depletion. H2B_1_29 1C.PEPAKSAPAPKKGSKKAVTKAQKKDGKKR and H2B_1_29 1H.PDPAKSAPAPKKGSKKAVTKAQKKDGKKR are more abundant in normal than in cancer cells upon treatment (Appendix A).

H3.3 histone variants—MGL-mediated methionine depletion does not induce statistically significant alterations on hPTMs in H3.3 histone variants (Appendix A).

### 3.3. Time Course of the Effect of MGL-Mediated Methionine Depletion on PTMs of Histone Variants in HT-29 Cancer Cells

H1 histone variants—24 and 48 h MGL treatments were applied to HT-29 cells in order to investigate how the deposition of marks on variants changed during progressive and sustained enzyme-mediated methionine starvation (Appendix A). Interestingly, incubation of HT-29 cells with MGL50 for 24 and 48 h induced a decrease of methylation marks, such as H1.4K25me2 and H1.2K33me1, and of the acetylation mark (H1.4S26ac), and H1.5 unmodified peptide. MGL50 treatment increased the relative abundance of the acetylation on K31 of H1.4 (Figure 1; Appendix A).

H2 histone variants—Variation in abundance of methylation at K5 and K15 of H2A1 displays opposite direction after 48 h treatment: the first one decreases and the second one increases. Little but significant variations were observed for H2A1K13ac and H2AZK7ac upon 24 h of methionine starvation, and for H2A3K15ac after 48 h (Figure 2 and Appendix A). H2AJ unmod exhibits a reduction, while H2AJK5acK9ac increases. At the same treatment time (24 h), the mono-methylation of H2AX presents a different pattern. We found an increase at K5 and a decrease at K9 induced by MGL50 (Figure 2; Appendix A).

In HT-29 cells upon treatment, the double acetylation at H2AZK4acK7ac and the triple acetylation at H2AZK4acK7acK11ac decrease after 48 h; on the contrary, the double acetylation at H2AZK7acK15ac decreases after 24 h (Figure 2; Appendix A).

We evaluated also the time course of alterations induced by MGL50 on hPTMs in H2B variants, upon 24 h of MGL-mediated methionine depletion. In this case, H2B1A unmod increases and H2B1AK86ac decreases compared to control. After 48 h, the relative abundance of H2B1B unmodified peptide decreases and the acetylation at Y83 increases (Figure 3; Appendix A).

H3 histone variants—hPTMs on the H3.3 histone variants when HT-29 cells were treated for 24 and 48 h are shown in Figure 4. The predominant mark in treated cancer cells is the H3.3 unmod. The methylation at K27 and K36 displays the same trend: the mono-methylation increases, and the di-methylation decreases upon 24 h of methionine starvation. We also observed a significant increase of the tri-methylation at K27 compared to control (Figure 4; Appendix A).

MGL50 treatment caused a higher acetylation at K27 and a lower one at S28 already after 24 h than the control. Subsequently, the S28ac is reduced, but not significantly (Figure 4; Appendix A).

The treatment for 48 h induced a reduction of K27me1K36me2, K27me1K36me1 and K27me3S28acK36me1 relative abundance (Figure 4; Appendix A).

The K27me3K36me1 and K27me1K36me3 increase already upon 24 h methionine depletion, but only the second one decreases significantly upon 48 h (Figure 4; Appendix A).

## 4. Discussion

In the last years, significant effort has been exerted to decipher the histone code and to establish a relationship between hPTMs and stress exposition. Histone code could be employed to manage genome/epigenome alterations and to establish a genetic/epigenetic crosstalk beyond DNA sequences. hPTMs play a pivotal role in many biological processes and represent the crucial mechanism by which histones exert regulatory control over different processes and therefore determine cellular identity and metabolism [48,49,50]. Histone variants, characterized by distinct amino acid sequences with respect to canonical histones, may alter nucleosome structure, stability, dynamics, and, ultimately, DNA accessibility to chromatin remodelers and modifiers. Histone variants are reported to be subjected to the same modifications as their canonical counterparts, but variant-specific modifications on residues that differ from their canonical counterparts have been documented. Histone variants are increasingly recognized as mediator of chromatin landscape diversification, but the functional significance of the incorporated marks has not been fully deciphered.

This study deepens and integrates our knowledge of hPTMs in HT-29 and Hs27 cells in absence and in presence of methionine depletion. In our previous work, we focused on histone 3 and 4 specific marks [6]. In this paper, we now characterize the hPTMs landscape of H1, H2A, H2B and H3.3 histone variants comparing a cancer cell line and a normal one, before and after treatment with MGL to induce methionine restriction. Methionine restriction is able to perturb methyl donor availability for the modification reactions and influence the metabolic status of cancer cells. Our mass spectrometry analysis points out that epigenetic marks deposited on histone variants are differently distributed in normal and cancer cells, emphasizing their essential role in normal cellular processes as well as in pathological signaling.

In HT-29 and Hs27cells epigenetic modification occurs mainly at lysine residues. Literature studies suggest that lysine acetylation is usually related to DNA damage response, cell-cycle control, chromatin architecture, RNA splicing, and transcription [51,52]. Indeed, histone methylation occurs on all basic residues (arginines, lysines and histidines) and takes part in gene repression or activation, depending on the sites and degree. In general, the turnover of methyl groups is believed to be slower than many other PTMs and histone methylation was originally thought to be irreversible [53].

In particular, in untreated cancer cells we observed several mono-, di- and tri-methylation (H1.4K25me3, H2A1K13me1, H2A3K15me1, H3.3K27me2, H3.3K27me3) on H1.4, H2A1, H2A3 and H3.3 histone variants, and a different pattern of mono-acetylation (H2A1K5ac, H2AZK15ac, H2A1K13ac, H2A1K15ac). Non-malignant cells show mono-methylation on H2A (H2A1, H2AZ and H2A3) and tri- or double-methylation on H3.3 histone variants. Nowadays, the relevance of PTMs modifications is well understood, but the complexity of the epigenetic signature and its impact on cellular context are unknown.

We carried out a detailed analysis of the histone PTMs induced by methionine starvation. Methionine is an essential amino acid and is involved in many biological processes, such as glutathione formation, polyamine synthesis, and methyl group donation. Our previous results highlighted that malignant cell growth was significantly impaired in methionine-depleted conditions, while normal cells growth was unaffected [6]. These effects could be attributed to the ability of normal cells to recycle homocysteine through methionine synthase to supply methionine endogenously. While this is true for normal cells, most of cancer cells lack the enzyme required to recycle homocysteine [4,17,54]. Therefore, methionine restriction significantly affects cancer viability without perturbing the survival of normal, healthy cells. This finding was the basis of methionine-restricted diet in murine models for cancer treatment and prevention strategies [17,55].

Our findings suggest that the changes of significant hPTM marks are different after MGL50 treatment in comparison with normal cells. These marks are abundant in cancer cells only after MGL50 administration. The most important alterations occur on H2A and H3.3 variants. The precise role of the histone modifications remains largely unclear and there is little information about the specific cellular mechanisms influenced by the deposition of such marks either in normal and cancer cells.

Using information deriving from in-silico tools such as the “HISTome2: The HISTone Infobase” [56] we tried to evidence the possible biological relevance of the observed modification.

Among the changes highlighted during methionine starvation induced by MGL treatment on the HT29 tumor cell line (Figure 5), changes involving histone H3.3 seem to play an important role. The histone variant H3.3 is considered a non-canonical variant of histone H3. Differently from canonical H3, expressed and incorporated into chromatin at S phase in a replication-dependent manner, variant H3.3 is expressed throughout the cell cycle, during S phase or in a replication-independent manner outside of S phase [47]. Although H3.3 is considered as a transcriptional activity mark [47,56], its functional importance was under debate. H3.3 is enriched at dynamic regions such as promoters, gene bodies and cis-regulatory elements; thus, serving as a mark of transcriptionally activated genes. Indeed, it is also deposited in transcriptionally inactive regions, including telomeres, pericentric heterochromatin and silent retroviral elements [47]. It was noted that mutation of H3.3K27 but not H3K27 results in an aberrant accumulation of pericentromeric transcripts and dysfunctional chromosome segregation [57,58]. Mutants of H3.3 at K27, G34, and K36 exhibit changes in their PTMs that modify chromatin and transcription and play important roles in tumorigenesis [59,60]. In this study, it was evidenced that both H3.3 lysine 27 (K27) and lysine 36 (K36) seem differently modified during methionine starvation. After 24 h treatment, a striking increase of methylation at K27 and K36 residues was observed. As H3.3 incorporation in euchromatin is largely associated with actively transcribed genes, thus the transient increase in methylation at K27 and K36 might be useful for the cell, under starvation stress, for reducing the transcriptional effort waiting for a restoration of the methionine levels. After a more prolonged period of starvation, 48h, it was seen a new decrease in the methylation pattern of both K27 and K36 that might be related to a change in transcription of cells related to a possible induction of programmed cell death pathways. The non-canonical histone H3.3 could be considered a new target for colorectal cancer therapy through methionine depletion, but more in-depth studies are needed.

As described in previous studies [50], our mass spectrometry approach provides a deep evaluation of the alteration induced by methionine starvation on PTM pattern both in normal and cancer cells. Based on the evidence accumulated by several studies [5,6,8,9,13,16,23,30], methionine restriction can become an additional cancer therapeutic strategy, in association with chemotherapy. Our data support this application, showing different epigenetic landscapes in normal and cancer cells under a starvation regimen. In addition, they provide the basis for identifying the role of not yet sufficiently studied histone modifications in response to cellular stresses such as lack of nutrients.

## 5. Conclusions

The analytical power of mass spectrometry was applied to the characterization of the PTMs of histone tails, the region more affected by epigenetic changes, including those caused by methionine depletion in cell cultures. Significant differences were observed between normal and cancer cells in the PTM pattern both in the absence and presence of MGL, an enzyme that is able to decrease the available methionine, suggesting diversified metabolic adaptations. Future studies are needed to examine the consequences of these profile changes in normal and cancer cells at the proteomic level.

## Figures and Tables

**Figure 1 cancers-15-00527-f001:**
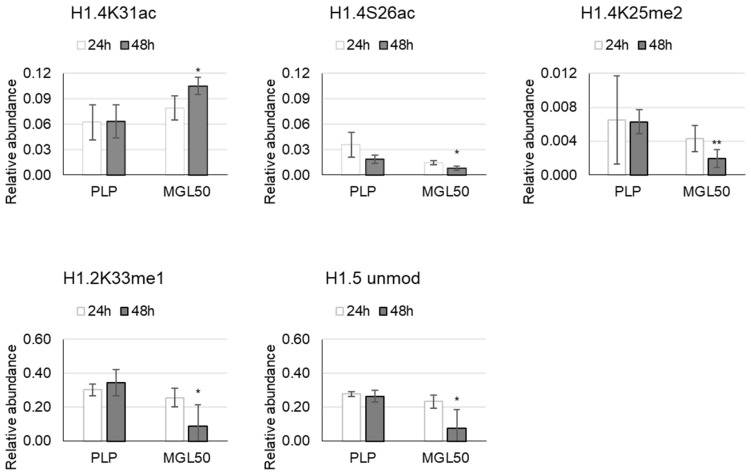
Effects of MGL50 on PTMs of H1 histone variant in HT-29 cells. Cells were seeded (2.5 × 10^5^ cells/well) into 12-well plates, treated with 0.03 mg/mL MGL (MGL50) for 24 and 48 h and then were collected. Histones were extracted and peptides were analyzed by mass spectrometry. Data are represented as mean ± standard deviation. PLP, control vehicle, cells treated with 0.5 mM PLP in PBS; MGL50, concentration of MGL able to reduce cell viability to 50%. Statistical analysis was carried out by comparing, at each incubation time, PLP vs MGL and reporting significant differences. Heteroscedastic two-tails *t*-test * *p* ≤ 0.05; ** *p* ≤ 0.01.

**Figure 2 cancers-15-00527-f002:**
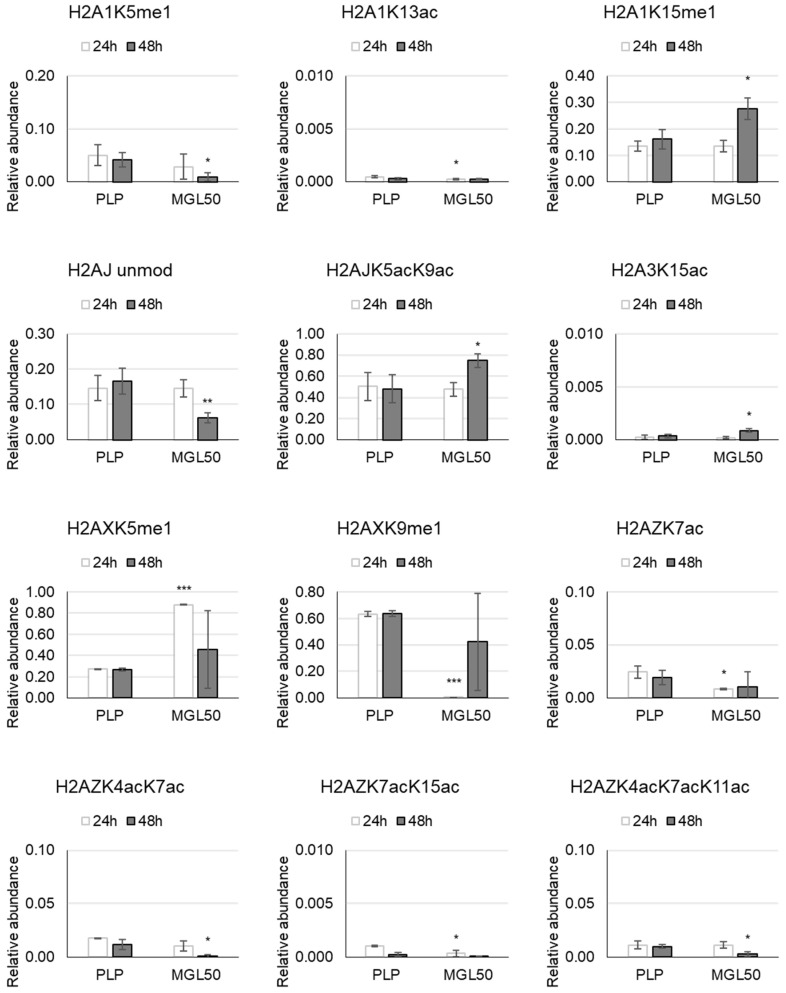
Effects of MGL50 on PTMs of H2A histone variant in HT-29 cells. Cells were seeded (2.5 × 10^5^ cells/well) into 12-well plates, treated with 0.03 mg/mL MGL (MGL50) for 24 and 48 h and then were collected. Histones were extracted and peptides were analyzed by mass spectrometry. Data are represented as mean ± standard deviation. PLP, control vehicle, cells treated with 0.5 mM PLP in PBS; MGL50, concentration of MGL able to reduce cell viability to 50%. Statistical analysis was carried out comparing, at each incubation time, PLP vs MGL reporting significant differences. Heteroscedastic two-tails *t*-test * *p* ≤ 0.05; ** *p* ≤ 0.01 and *** *p* ≤ 0.001.

**Figure 3 cancers-15-00527-f003:**
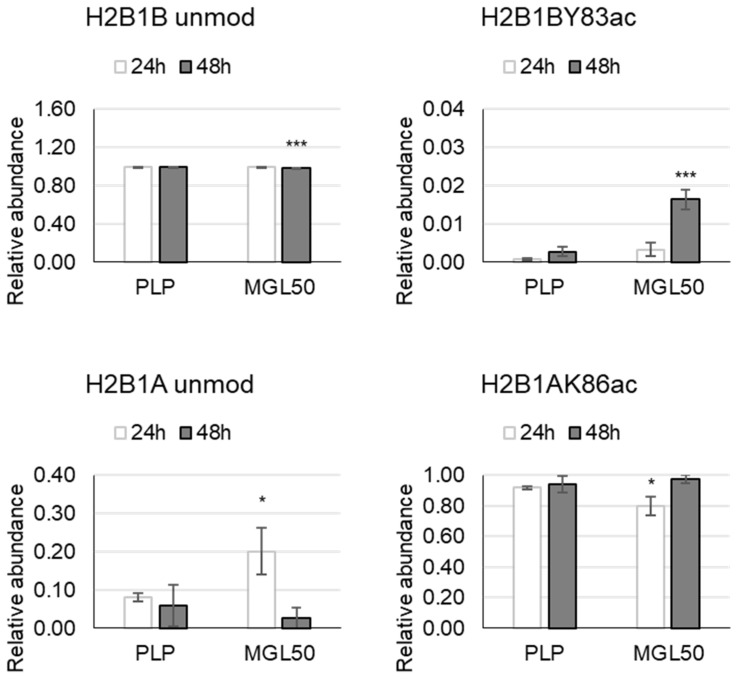
Effects of MGL50 on PTMs of H2B histone variant in HT-29 cells. Cells were seeded (2.5 × 10^5^ cells/well) into 12-well plates, treated with 0.03 mg/mL MGL (MGL50) for 24 and 48 h and then were collected. Histones were extracted and peptides were analyzed by mass spectrometry. Data are represented as mean ± standard deviation. PLP, control vehicle, cells treated with 0.5 mM PLP in PBS; MGL50, concentration of MGL able to reduce cell viability to 50%. Statistical analysis was carried out comparing, at each incubation time, PLP vs MGL reporting significant differences. Heteroscedastic two-tails *t*-test * *p* ≤ 0.05 and *** *p* ≤ 0.001.

**Figure 4 cancers-15-00527-f004:**
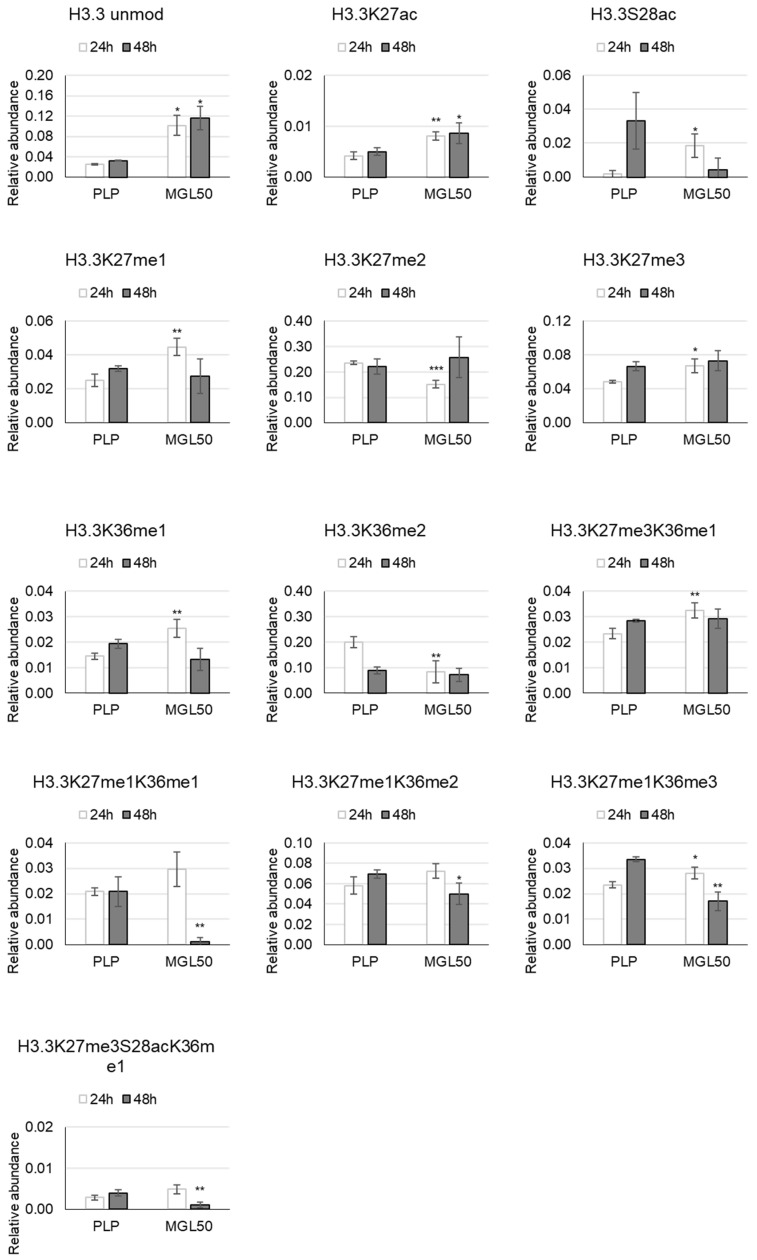
Effects of MGL50 on PTMs of H3.3 histone variant in HT-29 cells. Cells were seeded (2.5 × 10^5^ cells/well) into 12-well plates, treated with 0.03 mg/mL MGL (MGL50) for 24 and 48 h and then were collected. Histones were extracted and peptides were analyzed by mass spectrometry. Data are represented as mean ± standard deviation. PLP, control vehicle, cells treated with 0.5 mM PLP in PBS; MGL50, concentration of MGL able to reduce cell viability to 50%. Statistical analysis was carried out comparing, at each incubation time, PLP vs MGL reporting significant differences. Heteroscedastic two-tails *t*-test * *p* ≤ 0.05; ** *p* ≤ 0.01 and *** *p* ≤ 0.001.

**Figure 5 cancers-15-00527-f005:**
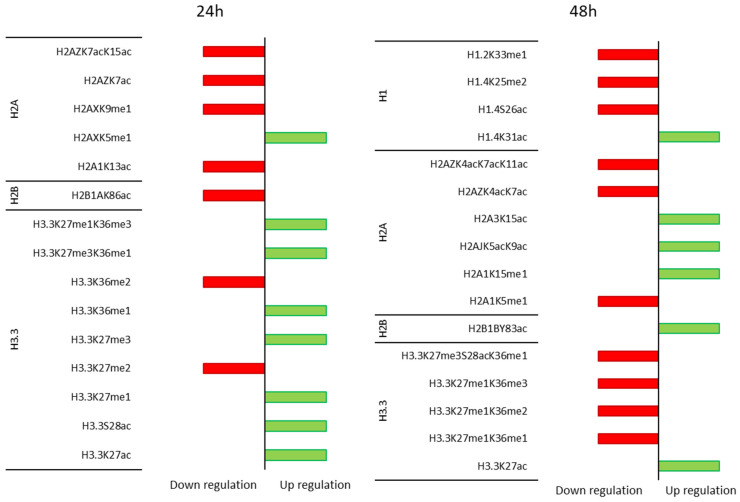
Leading PTMs in HT-29 cells after 24 and 48 h of MGL50 treatment. We reported in red the significant down-regulation and in green the significant up-regulation.

**Table 1 cancers-15-00527-t001:** PTMs differences of histone variants between HT-29 cancer cells and Hs27 non-malignant cells. Cells were seeded (2.5 × 10^5^ cells/well) into 12-well plates and collected after 48 h. Histones were extracted, and peptides were analyzed by mass spectrometry. Data are presented as mean and standard error and heteroscedastic two-tails *t*-test are applied. *p*-values are reported as −log2. In the last column, we highlighted in green statistically significant data obtained comparing normal vs. cancer cells.

hPMTs	Mean		Standard Error		*p*-Value
	Hs27	HT29	Hs27	HT29	
H14_25_32 K25me3	0.50%	1.62%	0.24%	0.12%	7.15
H2A1_4_11 unmod	73.43%	93.18%	2.46%	1.37%	9.95
H2A1_4_11 K5acK9ac	0.01%	0.22%	0.01%	0.05%	7.29
H2A1_4_11 K9me1	13.59%	0.98%	4.18%	0.80%	5.49
H2A1_4_11 K5me1	10.70%	2.60%	1.74%	0.39%	7.61
H2AJ_4_11 unmod	6.33%	19.01%	4.15%	0.51%	4.00
H2A1_1_11 unmod	50.90%	3.13%	9.77%	0.39%	5.23
H2A1_1_11 K5ac	47.64%	96.87%	9.83%	0.39%	5.30
H2AZ_1_19 K7ac	4.95%	2.68%	0.80%	0.44%	4.72
H2AZ_1_19 K15ac	0.01%	0.12%	0.01%	0.05%	4.17
H2A1_12_17 unmod	56.29%	48.09%	0.44%	2.58%	5.75
H2A1_12_17 K13ac	0.00%	0.02%	0.00%	0.01%	4.88
H2A1_12_17 K15ac	4.10%	18.99%	1.29%	6.49%	4.29
H2A1_12_17 K13me1	0.40%	2.50%	0.21%	0.72%	5.24
H2A3_12_17 unmod	5.73%	19.49%	1.62%	4.88%	5.03
H2A3_12_17 K15me1	14.10%	80.20%	2.36%	4.96%	12.98
H2A3_12_17 K13me1	80.18%	0.31%	3.97%	0.18%	9.27
H2A_1_88 H2A14s.HLQLAIR	44.43%	63.99%	8.29%	2.25%	4.34
H2A_1_88 H2AZ.AGGKAGKDSGKAKTKAVSR	53.41%	34.15%	7.96%	2.02%	4.46
H2B_1_29 1C.PEPAKSAPAPKKGSKKAVTKAQKKDGKKR	62.08%	46.65%	1.31%	5.79%	4.91
H2B_1_29 1H.PDPAKSAPAPKKGSKKAVTKAQKKDGKKR	11.11%	5.20%	0.84%	1.20%	7.01
H33_27_40 K27me2	4.87%	13.33%	3.20%	2.12%	4.21
H33_27_40 K27me3	0.74%	5.17%	0.91%	0.88%	6.28
H33_27_40 K36me3	4.09%	1.08%	0.92%	0.23%	5.83
H33_27_40 K27me1K36me2	13.23%	5.41%	2.30%	0.92%	5.79
H33_27_40 K27me1K36me3	4.62%	1.57%	0.90%	1.07%	4.17

## Data Availability

The data presented in the study are deposited in the Chorus repository (https://chorusproject.org/), accession number 2022, accessed on 4 March 2021.

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
