# Peer review of "Post-Translational Modifications of Histone Variants in the Absence and Presence of a Methionine-Depleting Enzyme in Normal and Cancer Cells"

_cancers, 2023, doi:10.3390/cancers15020527_

Round 1

Reviewer 1 Report

The manuscript by Montalbano et. al. describes the role of methionine in the post-translational modifications of histone variants in normal and cancer cells. Further, the author indicate a complex pattern of post-translational modifications on treatment with methionine gamma-lyase (MGL); which can be used to understand the molecular mechanism in MGL-based cancer therapy. 

The manuscript is well written and explained. I have no further comments on the manuscript.

Author Response

Response to reviewer 1

The reviewer wrote: the manuscript by Montalbano et. al. describes the role of methionine in the post-translational modifications of histone variants in normal and cancer cells. Further, the author indicate a complex pattern of post-translational modifications on treatment with methionine gamma-lyase (MGL); which can be used to understand the molecular mechanism in MGL-based cancer therapy.

The manuscript is well written and explained. I have no further comments on the manuscript.

We are grateful to the reviewer for his evaluation, and we are glad that our work was appreciated.

Reviewer 2 Report

Specific comments to the authors

The authors Serena Montalbano et al. of the submitted manuscript “Post-translational modifications of histone variants in the absence and presence of a methionine-depleting enzyme in normal and cancer cells” studies intensively the post-translational changes of histone variants (hPTMs) in normal and cancer cells by mass-spectrometry in relation to methionine gamma-lyase (MGL). Therefore, the authors applied mass spectrometry after treatment with MGL and PTM preparation in combination with the online software tool EpiProfile 2.0 in-vitro (using the human colon adenocarcinoma cell line HT-29 and the human skin fibroblast cell line Hs27).

In summary, based on their investigations program the authors found complex and heterogeneous pattern of PTMs on histone variants and striking differences between normal and cancer cells in comparison to MGL treatment. Therefore, the authors suggested that investigation of post-translational changes of histone variants after methionine depletion could help to understand the molecular mechanisms induced by methionine starvation causing cancer cell death.

Overall, the manuscript give a survey of possible PTMS in dependency of MGL treatment and in comparison of a normal control and a cancer cell line. The manuscript (including presentation) is mostly comprehensible and convincing. The methods are mostly well described. Although the results and discussion are clearly presented, the authors (see specific comments) must perform some minor to major changes (statistical and in-silico-analysis) to improve the manuscript. In conclusion, the presented data have some interesting aspects. After incorporating the mentioned specific comments (see below) the manuscript has the potency to be accepted.

Specific comments

Abstract: The abstract contains only vague results, which should be definitively specified by the authors in more details. Furthermore, the conclusion of the abstract is largely speculative, too. Please use correct the abbreviations (PMT instead of PTM) throughout the manuscript. The authors wrote, that the findings “will help to decipher the epigenetic modifications associated with methionine depletion, deepening the understanding of the molecular mechanisms induced by methionine starvation causing cancer cell death” but now cancer cell death related investigations are presented. Please clarify.

Material and Methods: The MGL catalytic activity should be shown by the authors in a supplementary figure. Please explain the selection of the applied human cancer cell lines. The correct reference of EpiProfile 2.0 is “J Proteome Res. 2018 Jul 6;17(7):2533-2541” (see PMID: 29790754). Please specify the used statistical program. Furthermore, please perform statistical tests for the data distribution before using the mentioned T-test.

Results:

# Table 1: Please describe the meanings of the used colors. The authors should apply hierarchical cluster analysis to define subgroups of the PTMs and to indicate positive and negative interactions, too.

# Figure 1 to 4: The authors should compare the findings in an additional table or figure to show the “leading” PTMs in dependency of time and treatment. As the authors stated, that MGL treatment reduce the cell viability, this is not shown by experiments. Please clarify adequately.

Finally, the biological relevance of PTMs should be analyzed in more details using other in-silico tools (see Epigenetics Chromatin, 2020 Aug 3;13(1):31 PMID: 32746900).

Discussion: Please discuss the findings in relation to biological relevance to rule ut the definitive impact of the findings.

Author Response

Response to reviewer 2

Specific comments to the authors

The authors Serena Montalbano et al. of the submitted manuscript “Post-translational modifications of histone variants in the absence and presence of a methionine-depleting enzyme in normal and cancer cells” studies intensively the post-translational changes of histone variants (hPTMs) in normal and cancer cells by mass-spectrometry in relation to methionine gamma-lyase (MGL). Therefore, the authors applied mass spectrometry after treatment with MGL and PTM preparation in combination with the online software tool EpiProfile 2.0 in-vitro (using the human colon adenocarcinoma cell line HT-29 and the human skin fibroblast cell line Hs27).

In summary, based on their investigations program the authors found complex and heterogeneous pattern of PTMs on histone variants and striking differences between normal and cancer cells in comparison to MGL treatment. Therefore, the authors suggested that investigation of post-translational changes of histone variants after methionine depletion could help to understand the molecular mechanisms induced by methionine starvation causing cancer cell death.

Overall, the manuscript give a survey of possible PTMS in dependency of MGL treatment and in comparison of a normal control and a cancer cell line. The manuscript (including presentation) is mostly comprehensible and convincing. The methods are mostly well described. Although the results and discussion are clearly presented, the authors (see specific comments) must perform some minor to major changes (statistical and in-silico-analysis) to improve the manuscript. In conclusion, the presented data have some interesting aspects. After incorporating the mentioned specific comments (see below) the manuscript has the potency to be accepted.

Specific comments

Abstract: The abstract contains only vague results, which should be definitively specified by the authors in more details. Furthermore, the conclusion of the abstract is largely speculative, too. Please use correct the abbreviations (PMT instead of PTM) throughout the manuscript. The authors wrote, that the findings “will help to decipher the epigenetic modifications associated with methionine depletion, deepening the understanding of the molecular mechanisms induced by methionine starvation causing cancer cell death” but now cancer cell death related investigations are presented. Please clarify.

We are very grateful for the time that the Reviewer spent for revising our manuscript. We checked and clarified the abstract consequently. Please, find below the update abstract version as present in the newer version of our manuscript. In addition, we paid attention and corrected the abbreviations (PTM) in the draft.

Methionine is an essential amino acid involved in the formation of polyamines and a precursor metabolite for DNA and protein methylation. The dependence of cancer cells on methionine has triggered extensive investigations aimed at its targeting for cancer therapy, including the exploitation as a therapeutic tool of methionine γ-lyase (MGL), a bacterial enzyme that degrades methionine, capable of inhibiting cancer cells growth due to methionine starvation. We have exploited the high-resolution power of mass spectrometry to compare the effects of reduced availability of the methyl donor SAM, induced by MGL treatment, on the post-translational modifications of the histone tails in normal Hs27 and cancer HT-29 cells. In the absence of MGL, our analysis detected a three-fold higher relative abundance of trimethylated K25 of H1.4 in HT-29 than Hs27 cells, and a complex pattern of methylated, unmethylated and acetylated peptides in H2 and H3.3. In the presence of MGL, in HT-29, the peptide H2A1_4_11 is predominantly unmodified with mono-methylated K5 increasing upon treatment, whereas in Hs27 cells, H2A1_4_11 is monomethylated at K5 and K9 with these marks decreasing upon treatment. The time dependence of the effects of MGL-mediated methionine depletion on PTMs of histone variants in HT-29 cancer cells was also monitored. Overall, our present data on histone variants H1, H2A, H2B as well as H3.3 integrated with our previous studies on histones H3 and H4, shed light on the epigenetic modifications associated with methionine starvation and associated cancer cell death.

Material and Methods: The MGL catalytic activity should be shown by the authors in a supplementary figure. Please explain the selection of the applied human cancer cell lines. The correct reference of EpiProfile 2.0 is “J Proteome Res. 2018 Jul 6;17(7):2533-2541” (see PMID: 29790754). Please specify the used statistical program. Furthermore, please perform statistical tests for the data distribution before using the mentioned T-test. 43.

As correctly highlighted by the Reviewer, we added in “Material and Methods” the information related to MGL catalytic activity. We reported chemical reaction for methionine degradation and our references for previous experiments and results.

MGL degrades methionine according to the reaction: L-methionine + H2O → methanethiol + NH3 + α-ketobutyrate. MGL catalytic activity was determined using L-methionine as a substrate and measuring the rate of α-ketobutyrate production in the coupled reaction with D-2-hydroxyisocaproate dehydrogenase (HO-HxoDH) monitoring the decrease of NADH absorption at 340 nm (Δε=6220 M-1 cm-1) at 37 °C. The rate of α-ketobutyric acid production was measured in a coupled assay at 30 °C. The reaction mixture contained 100 mM potassium phosphate, 100 mM PLP, 5 mM DTT, 0.2 mM NADH, pH 7.2. The amount of the enzyme that catalyzes the formation of 1.0 μmol min−1 of α-ketobutyrate at pH 8.0, 37 °C, was defined as one unit of enzyme activity. Kinetic parameters of MGL mutants in the α,γ-elimination reaction with L-methionine were previously reported to be kcat= 7.9 s-1 ±0.3; KM = 0.8 mM ±0.1; kcat/KM = 9.7x103 M-1s-1 (Raboni et al. Biochim Biophys Acta Proteins Proteom. 2018).

Regarding the selection of the cell line, we now specified in the manuscript that the cell line HT-29 was selected after a preliminary screening of MGL dependence against a panel of different cancer cells. HT-29 showed a very interesting antiproliferative profile after MGL50 treatment, as reported in our previous publication.

In a previous study (Raboni et al. Front Mol Biosci. 2021), we analyzed the effects of methionine depletion caused by MGL treatment on cancer (HT-29) and normal cells (Hs27) and documented the higher susceptibility of HT-29 cells with respect to Hs27 non-malignant cells. The concentration inhibiting HT-29 cell proliferation at 50% (MGL50) was 30 µg/ml at 72 h of incubation, to be compared with more than 460 µg/ml for Hs27.

Finally, thank you for noticing the issue with citing EpiProfile 2.0 and the proper statistical analysis. We have corrected the citation and added clarification to the analysis.

Histone peptide data were analyzed using EpiProfile 2.0 software [citation]. The raw abundance of each (un)modified peptide was obtained by performing extracted ion chromatography for each peptidoform. To obtain peptide relative abundance, and thus relative abundance of histone PTMs, the sum of all different modified forms of a histone peptide was considered as 100% and the area of the particular peptide was divided by the total area for that histone peptide in all of its modified forms. The relative ratio of two isobaric forms was estimated by averaging the ratio for each fragment ion with different mass between the two species. The resulting peptide lists generated by EpiProfile were exported to Microsoft Excel for statistical analysis. Due to the relatively limited number of replicates (N=3, typical of experiments involving cell lines), data distribution could not be tested using e.g. the Shapiro-Wilk test. For this reason, data distribution of data points across replicates was assumed to be normal due to the nature of the sample, but it was not formally tested. To minimize the probability of false positives, we used a t-test heteroscedastic with two-tails (significant if p<0.05).

Results:

# Table 1: Please describe the meanings of the used colors. The authors should apply hierarchical cluster analysis to define subgroups of the PTMs and to indicate positive and negative interactions, too.

We deleted the color code in Table 1. We thank the Reviewer for having underlined this issue.

# Figure 1 to 4: The authors should compare the findings in an additional table or figure to show the “leading” PTMs in dependency of time and treatment. As the authors stated, that MGL treatment reduce the cell viability, this is not shown by experiments. Please clarify adequately.

As suggested by the Reviewer, we reported an additional table to show the “leading” PTMs (Figure 5). We chose the most important PTMs observed after 24h and 48h of treatment in HT-29 cells (see below).

As per the effect of MGL50 treatment in Hs27 and HT-29 cells, we highlighted in the first paragraph of the result section our previous publications where these effects were characterized in detail [Raboni et al. 2018; DOI: 10.1016/j.bbapap.2018.09.011; Raboni et al. 2021; doi:

(from the result section): In a previous study [6], we analyzed the effects of methionine depletion caused by MGL treatment on cancer (HT-29) and normal cells (Hs27) and documented the higher susceptibility of HT-29 cells with respect to Hs27 non-malignant cells. The concentration inhibiting HT-29 cell proliferation at 50% (MGL50) was 30 µg/ml at 72 h of incubation, to be compared with more than 460 µg/ml for Hs27.

Finally, the biological relevance of PTMs should be analyzed in more details using other in-silico tools (see Epigenetics Chromatin, 2020 Aug 3;13(1):31 PMID: 32746900).

Discussion: Please discuss the findings in relation to biological relevance to rule ut the definitive impact of the findings.

We thank the reviewer for his suggestions which allowed us to try to deepen the biological role of some histone modifications. In particular, we have added in the discussion a part concerning the modifications of the non-canonical histone H3.3. We found this to be one of the most innovative findings of our work since our pipeline allowed us to discriminate the relative quantification of histone modifications from highly similar histone variants. Regarding the regulation of other modifications such as H2A methylation or H1 and H2B acetylation, we are not comfortable speculating on their biological impact on the cell, as there are still lots of unknown regarding their function. We tried to be as transparent as possible with these aspects (below).

The precise role of the histone modifications remains largely unclear and there is little information about the specific cellular mechanisms influenced by the deposition of such marks either in normal and cancer cells. Using information deriving from in-silico tools such as the “HISTome2: The HISTone Infobase”(59) we tried to evidence the possible biological relevance of the observed modification  Among the changes highlighted during methionine starvation induced by MGL treatment on the HT29 tumor cell line (Figure 5), changes involving histone H3.3 seem to play an important role. The histone variant H3.3 is considered a non-canonical variant of histone H3. Differently from canonical H3, expressed and incorporated into chromatin at S phase in a replication-dependent manner, variant H3.3 is expressed throughout the cell cycle, during S phase or in a replication-independent manner outside of S phase [48]. Although H3.3 is considered as a transcriptional activity mark [48,59], its functional importance was under debate. H3.3 is enriched at dynamic regions such as promoters, gene bodies and cis-regulatory elements; thus, serving as a mark of transcriptionally activated genes. Indeed, it is also deposited in transcriptionally inactive regions, including telomeres, pericentric heterochromatin and silent retroviral elements [48]. It was noted that mutation of H3.3K27 but not H3K27 results in an aberrant accumulation of pericentromeric transcripts and dysfunctional chromosome segregation [60-61]. Mutants of H3.3 at K27, G34, and K36 exhibit changes in their PTMs that modify chromatin and transcription and play important roles in tumorigenesis [62-63]. In this study, it was evidenced that both H3.3 lysine 27 (K27) and lysine 36 (K36) seem differently modified by during methionine starvation. After 24h treatment, a striking increase of methylation at K27 and K36 residues was observed. As H3.3 incorporation in euchromatin is largely associated with actively transcribed genes, thus the transient increase in methylation at K27 and K36 might be useful for the cell, under starvation stress, for reducing the transcriptional effort waiting for a restoration of the methionine levels. After a more prolonged period of starvation, 48h, it was seen a new decrease in the methylation pattern of both K27 and K36 that might be related to a change in transcription of cells related to a possible induction of programmed cell death pathways. The non-canonical histone H3.3 could be considered a new target for colorectal cancer therapy through methionine depletion, but more in-depth studies are needed.

Round 2

Reviewer 2 Report

Specific comments to the authors

In the revised version of the manuscript, the authors were able to address the previously mentioned concerns in a very appropriate and convincing manner. Therefore, the revised manuscript "Post-translational modifications of histone variants in the absence and presence of a methionine-depleting enzyme in normal and cancer cells" should be accepted in its current form.